# Climate Smart Agriculture Implementation on Coffee Smallholders in Indonesia and Strategy to Accelerate

**Fadjry Djufry [1], Suci Wulandari [2,* ] and Renato Villano [3]**

[1] Indonesian Agency for Agricultural Research and Development, Jakarta 12540, Indonesia; fadrydjufry@pertanian.go.id

[2] National Research and Innovation Agency, Jakarta, 12710, Indonesia

[3] UNE Business School, University of New England, Armidale, NSW 2351, Australia; rvillan2@une.edu.au

\* Correspondence: suci021@brin.go.id

**Abstract:** Sustainable coffee production is significantly threatened by climate change. While implementing CSA practices offers numerous benefits, adoption rates remain low. Coffee plantations are dominated by smallholders and located in rural areas, making them more complex and requiring a comprehensive analysis and intervention. This study used an exploratory approach to assess farmers' preferences for CSA practices, identify barriers to implement, and design a support system model. The investigation focused on Arabica and Robusta farmers, with case studies from two Indonesian production centres. Preferences assessment used conjoint analysis, barriers evaluation used Mann–Whitney analysis, model development used synthetic approaches, and priority analysis used the Analytical Hierarchy Process. The study revealed that diversification is more desirable than cultivation, soil management, and water management. Arabica farmers preferred intercropping with annual crops, whereas Robusta farmers preferred perennials crops. Robusta farmers assessed that agricultural inputs, such as labor, capital, climatic data, and farm equipment and machinery, existed as barriers. However, these represent a lesser issue for Arabica farmers. We proposed agricultural innovation support system, consisting of innovation support facilities and services, as a comprehensive support system model to accelerate CSA implementation. Further analysis showed that the priority strategy for Arabica farmers is support services that focus on network development, while for Robusta farmers is support facilities that focus on climate information system development.

**Keywords:** climate change; smallholders; climate-smart; innovation; support system; coffee

## 1. Introduction

Increasing temperature by 1.5 °C as a result of global warming has caused unexpected extreme weather events and generated severe negative impacts on human livelihoods and environments [1]. In particular, climatic variables have a robust exert significant on agricultural productivity growth [2]. Coffee, one of the most heavily traded agricultural commodities on a global scale, has been identified as a very vulnerable plant species to climate change [3]. The disruption of growth driven by climate change poses substantial issues for coffee plants, resulting in a decline in production and quality [4].

Production of Arabica coffee (*Coffea arabica*) and Robusta coffee (*Coffea canephora*) is mostly reliant on the optimum rainfall and temperature [5], nevertheless, an increase in severity and frequency of climate disturbance has an impact on coffee production [6]. The main climate constraints for coffee production are drought and unfavourable temperatures [7]. Groundwater availability at various stages of coffee plant growth significantly affects coffee production [8]. As a result of increasing temperatures and changing precipitation patterns, coffee plants are experiencing decreased yields, decreased quality, and induce the occurrence of pests and diseases [9]. Temperature changes will directly impact the climatic suitability of coffee growing areas [10]. In the year 2050, climate change will

reduce the amount of suitable land for Arabica in Indonesia by 67% and may create an additional 28% of suitable land [11]. Along with reducing yield and quality, climate change increases the cost of coffee production [12].

Climate-Smart Agriculture (CSA) is the workable alternative to address climate change. The CSA is an integrated governance framework for agricultural methods and technologies that increase crop productivity while increasing climate resilience and reducing greenhouse gas emissions [13]. Climate-smart technologies are highly diverse, with more than 1700 unique combinations of production systems, regions, and technologies as CSA practices [14]. A range of CSA practices is implemented as several entry points for climate change adaptation and mitigation, such as soil management, water management, chemical input management, and farm diversification [15]. Others CSA practices are affected by the transition from industrial techniques to data-driven management and automation [16]. Those are significantly influenced by process automation, data analysis and processing, as well as farm operations control and management [17]. New technologies such as the Internet of Things and cloud computing are expected to accelerate agricultural development by introducing more robots and artificial intelligence in farming and big data utilisation [18].

Indonesia is vulnerable to the adverse effects of climate change and contributes to global greenhouse gas emissions [19]. It has a tropical monsoon climate characterised by modest seasonal and temperature variations, a lack of wind, high humidity, and periodic rainfall [20]. Global warming causes climate changes that cannot be reformed in a short time, therefore, overcoming them requires systematic policies and sustainable agricultural practices [21]. Indonesia has published the Long-Term Strategy for Low Carbon and Climate Resilience 2050, which intends to contribute to the achievement of global goals and national development objectives by finding a balance between emission reduction, economic growth, justice, and the development of climate resilience. Indonesia recognises that mitigation and adaptation provide complementary roles in responding to climate change at diverse geographical, temporal, and institutional scales [19].

Implementing CSA in coffee farming systems significantly impacts coffee production and contributes to carbon sequestration on a global scale [22]. CSA provides agricultural production practices that benefit farmers through increased productivity and profitability and reduced vulnerability to climate change [15]. Despite the fact that CSA provides numerous benefits and technological innovation is highlighted as playing an important role, it is not always adopted [23]. This phenomenon also happens among coffee farmers who pay less attention to technologies to increase productivity [24]. Coffee farmers are dominated by smallholder plantations within the Indonesian context, reaching 95.45% [25]. Implementation of CSA practices at the smallholder level faces various barriers related to the character of farmer and farming system, such as shortage of agricultural land, land tenure issues, lack of adequate knowledge of CSA, slow return on investment, and inadequate policy and implementation schemes [26]. Anyway, CSA adoption also is related to technological characteristics and external factors, such as market access and traditional culture [27].

The lack of research on climate variability and the impact of climate change on coffee production in Indonesia presents a challenge for implementing CSA [28]. Studies on the preferences for CSA practices in coffee smallholders that differentiate between Arabica and Robusta coffee have not been conducted yet. Given the relative importance of coffee in the livelihoods of smallholders and the differences associated with the types of coffee, it is imperative to evaluate the farmer's preferences for different CSA practices and map the barriers to implementation by differentiating into Arabica farmers and Robusta farmers. Information on farmer preferences will accelerate the implementation of conventional CSA and build a foundation for implementing cutting-edge technologies such as the Internet of Things, artificial intelligence, and robotics.

While CSA practices have been shown to increase agricultural systems' resilience to climate change, they are not components that can be superimposed without considering the entire system [29]. The state support system for agriculture is essential for sustainable

agricultural development [30]. Effective targeted institutions and policies are needed to reduce resource constraints that hinder farmers' capacity to adopt CSA [31]. A well-designed and supported innovation system can facilitate the transition to sustainable agriculture, following different approaches and paradigms [32]. As a result, this research will include developing a support system model to enable the implementation of CSA.

Therefore, this study aims to (1) analyse farmers' preferences towards CSA at Arabica and Robusta coffee production centres in Indonesia, (2) analyse barriers to implementing CSA at the farmer's level, and (3) provide a model of agricultural innovation support system to accelerate CSA implementation on coffee smallholders. Implementing CSA practices while considering support system development will result in a more sustainable coffee production system.

## 2. Study Design and Methods

### 2.1. Study Area and Respondents

The research was conducted in West Java Province for the centres of Arabica coffee production and Bengkulu Province for Robusta coffee production. The study focused on smallholder plantations, which comprise small business and household business, per the BPS-Statistic Indonesia [33]. A coffee plantation small business is established or operated commercially by an individual company without a notary deed but meets the criteria of having an area of at least 1 hectare or 1250 trees. A coffee plantation household business is neither a legal entity nor organised or managed by households; it has also not met the criteria for small business.

Data were collected in 2020 in Bandung Regency in West Java Province and Kepahiang and Rejang Lebong Regencies in Bengkulu Province (Figure 1). As shown in Table 1, 205 coffee farmers in Arabica and Robusta coffee production centres were interviewed.

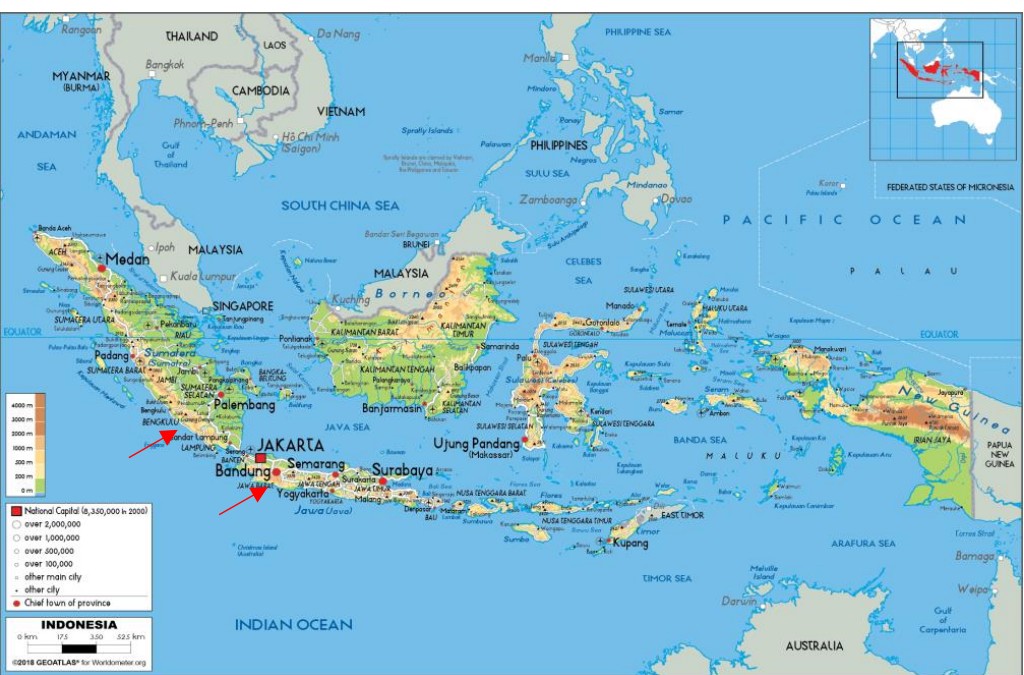

**Figure 1.** The study areas.

**Table 1.** Location and the number of respondents.

| Respondent Groups | Location | | Number of Respondents |
|---|---|---|---|
| | Province | Regency | |
| Arabica coffee smallholders | West Java | Bandung | 79 |
| Robusta coffee smallholders | Bengkulu | 1. Rejang Lebong<br>2. Kepahiang | 50<br>76 |
| Total respondents | | | 205 |

The sample farmers were selected through a two-stage sampling approach. In the first stage, the purposive sampling method was used to choose the sample provinces, followed by the regencies as the study site. In the second stage, random sampling was used to select respondents, and the sample size was carried out proportionally by considering the number of national Arabica and Robusta coffee farmers.

*2.2. Data Type and Collection Method*

This paper examined primary and secondary sources of data. Primary data include farming unit characteristics, farmer preferences for CSA practices, barriers to CSA implementation, and priority strategies for accelerating CSA implementation. Primary data were obtained through farmers' interviews using pre-tested and designed questionnaires, field observations, and interviews with resource persons. Secondary data include site descriptions, regional and national coffee production, and area statistics. The main sources of secondary data are The Ministry of Agriculture and Central Statistics Agency.

The questionnaires included modules of (1) respondent profiles and farming systems characteristics, (2) farmer preferences for CSA practices, and (3) barriers faced by farmers. Closed-ended questions were used in the module on respondent profiles and farming system characteristics. In farmer preferences for the CSA practices module, farmers evaluated CSA practices on a scale from 1 to 4, with "1" indicating definitely not, "2" indicating probably not, "3" indicating probably yes, and "4" indicating definitely yes. In the barriers faced by farmers modules, the questions assess barriers faced by using a Likert scale with "1" indicating strongly disagree, "2" indicating disagree, "3" indicating neutral, "4" indicating agree, and "5" indicating strongly agree.

*2.3. Data Analysis Method*

This research used a multi-method design, divided into the following stages: (1) identify CSA practices and analysis of farmer preferences, (2) analysis of barriers faced by farmers, (3) develop a conceptual model as a guide for formulating strategies, and (4) priority analysis of strategies (Figure 2). The output is a recommendation for accelerating CSA implementation, including technical and institutional recommendations.

The analysis used conjoint analysis to determine farmers' preferences for CSA practices. Several research studies have utilised this method, such as farmers' preferences for CSA [34] and farmers' preferences for coffee certification [35]. It can determine preferences by evaluating various attributes' usefulness and relative importance [36]. Data can be generated through a survey, in which respondents are asked to rate choices based on their level of importance or preference [37]. Several steps for assessing a farmer's preference for CSA practices consist of:

1.  Problem formulation. This stage aims to identify the attributes and levels of CSA practices based on the literature and confirmed by field observations.
2.  Formation of stimuli. This stage aims to reduce the combination. With the orthogonal design using SPSS 26, the conjoint analysis generated 9 stimuli. The orthogonal design was used to ensure the reasonable number of stimuli the respondents evaluated.
3.  Determination of data type required. The data used were numerical numbers 1–4, describing the farmers' preference for stimuli.

4.　Selection of conjoint analysis procedure and analysis using [38]:

$$U(x) = \sum_{i=1}^{m_i} \sum_{j=1}^{k_j} a_{ij} x_{ij} \tag{1}$$

Description:

U(x) = overall utility of an alternative;
$a_{ij}$ = part worth or utility contribution attribute i from j level;
$k_i$ = number of attribute level i;
$m_j$ = number of attributes level j;
i = 1,2, . . . , m (attribute i);
j = 1,2, . . . , k (level j);
$x_{ij}$ = dummy variable 1 = yes, 0 = no.

5.　Interpretation of results: The higher utility value level is the preferred level. The total utility value for each combination is the same as the total utility value for each attribute level.

6.　Test the reliability and validity of the results. The accuracy test of the results is determined by looking at the value of Kendall's Tau correlation between the conjoint analysis results and respondents' actual opinions.

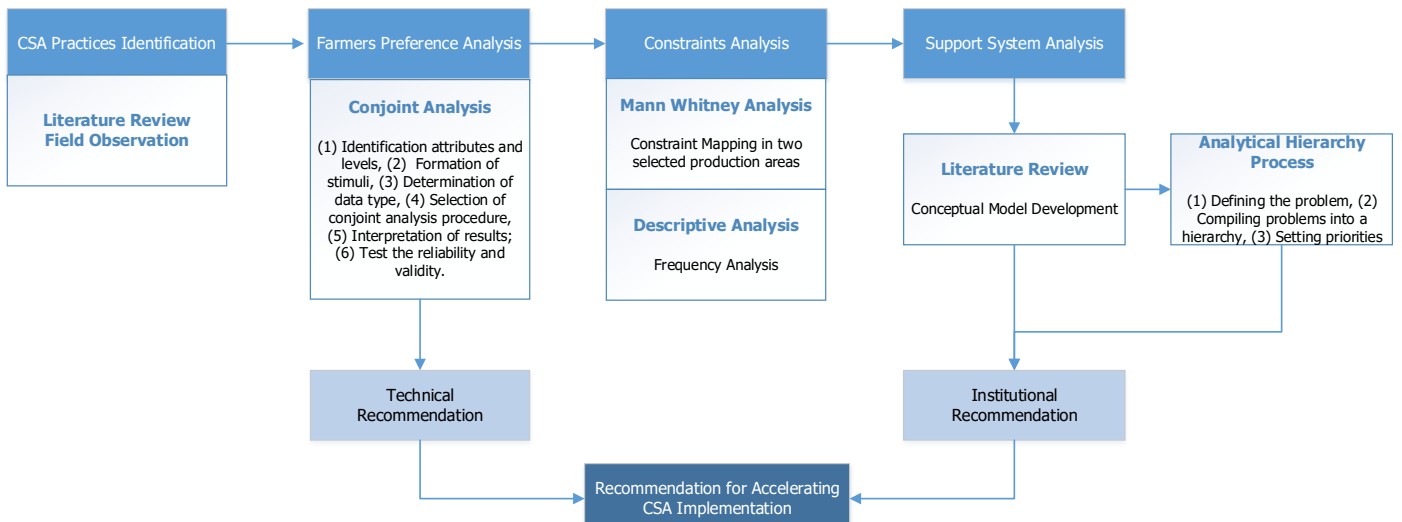

**Figure 2.** Research stages and data analysis method.

The following analysis examines the barriers faced by farmers using Mann–Whitney analysis, a type of non-parametric statistical testing used to compare the distributions of two independent populations. It can be used as a substitute for the t-test for independent samples in cases where the sample's value does not follow a normal distribution or the distribution of values is unknown [39]. It states whether the difference is significant or coincidental but does not explain why the difference exists [40]. Therefore, in this study, the analysis is complemented by other analyses. The significance threshold is set at 0.05, and the formula is shown below:

$$U_1 = n_1 n_2 + \frac{n_1 (n_1 + 1)}{2} - \sum R_1 \tag{2}$$

$$U_2 = n_1 n_2 + \frac{n_2 (n_2 + 1)}{2} - \sum R_2 \tag{3}$$

Description:

U = statistic test of Ui;
$n_1$ = number of group 1;
$n_2$ = number of group 2;
$\sum R_1$ = sum of the ranks for group 1;
$\sum R_2$ = sum of the ranks for group 2.

The necessary support system model was built after assessing the barriers faced. A conceptual support system model was developed based on the literature review and considering the conditions in the field. AHP will then do a prioritising analysis utilising the support system's conceptual model using Expert Choice software. AHP is a multi-criteria decision-making tool that can be used to define priorities and ratings [41]. The AHP steps consist of: (1) defining the problem and setting goals, (2) compiling problems into a hierarchy, and (3) setting priorities for each problem element in the hierarchy.

The basic concept of AHP is using a pairwise comparison matrix to generate the relative weights between the criteria and alternatives. The recommended values for creating a pairwise comparison matrix are 1 = equal important, 3 = slightly important, 5 = strongly important, 7 = very strong important, and 9 = extremely important. In addition to the values above, intermediate values (2, 4, 6, and 8) can also be used. If the interests are reversed, we can use reciprocal numbers. The analysis was conducted by extracting data and information from farmer groups, local governments, and researchers.

## 3. Results and Discussion

### 3.1. Impact of Climate Change on Coffee Plantation

The requirements for growing Arabica coffee plants are 1000–2000 m above sea level, rainfall of 1250–2500 mm/year, dry months (rainfall <60 mm/month) of 1–3 months, and an average air temperature is 15–25 °C. Bandung Regency, an Arabica coffee production centre, is surrounded by mountains and hills. Bandung Regency has a tropical climate influenced by monsoons, with annual rainfall ranging from 1500 to 4000 mm/month. Temperatures range between 12 °C and 24 °C, with humidity levels between 78 and 70% during the rainy season and 70% during the dry season. Coffee in Bandung Regency is grown more than 1000 m above sea level. The area of Arabica coffee in Bandung Regency is 7463 ha with 6667 tons and involves 15,366 farmers [42].

Robusta coffee is planted at an altitude of 100–600 m above sea level, with rainfall of 1250–2500 mm/year and dry months (rainfall <60 mm/month) for three months, and the air temperature is 21–24 °C. Rejang Lebong and Kepahiang Regency are the primary coffee producers in Bengkulu province, with a dominance of the Robusta coffee. Rejang Lebong Regency is a hilly area with an altitude of 100 to 1000 m above sea level and flat to bumpy slopes. The average rainfall is 233.75 mm/month, with an average number of rainy days of 14.6 days/month in the dry season and 23.2 days/month in the rainy season. The average temperature is 17.73–30.94 °C, with an average relative humidity of 85.5%. Kepahiang Regency consists of a highland area with an altitude of 350 m to more than 1200 m above sea level. Kepahiang has a tropical climate with an average rainfall of 233.5 mm/month—three dry months and nine wet months. The average relative humidity is 85.21%, and the average daily temperature is 23.87 °C, with a maximum temperature of 29.87 °C and minimum temperature of 19.65 °C. The areas of Robusta coffee in Rejang Lebong and Kepahiang Districts are 19,572 and 23,566 ha, with 15,740 and 19,204 tonnes of production, involving 18,475 and 13,953 farmers [42].

Weather condition is one of the factors that affect the quality of coffee, along with plant types, topography, and the management provided during the planting, harvesting, storage, export preparation, and transportation periods [43]. Based on data from 91 observation stations in Indonesia, Indonesia's normal air temperature for the 1981–2010 period was 26.6 °C, and the average air temperature in 2020 was 27.3 °C. The year 2016, with an anomaly value of 0.8 °C, was the warmest, and the year 2020 placing second with an anomaly value of 0.7 °C.

Global warming affects coffee production by changing rainfall patterns and temperature, making areas unsuitable for production. The distribution of rainfall and air temperature disrupts plant phenology, thus negatively impacting their productivity and quality. The changing climatic conditions facilitate the spread of pests and diseases [44]. The increase in pest attacks causes a decrease in coffee beans' quality or even damage yields and crops [45].

Arabica coffee needs more thorough handling. Temperature changes have resulted in the suboptimal production and quality of Arabica coffee. Rising temperatures and changing rainfall patterns are expected to severely reduce Indonesia's total area of climatically suitable Arabica coffee-growing region by 2050 [46]. Robusta coffee trees can grow at low altitudes and in hot climates with few water conditions. However, weather variability has affected production and quality. Robusta coffee tends to be more resistant to pests and diseases. In 2017, coffee farmers in Kepahiang Regency, Bengkulu Province, experienced harvest failures due to the very high intensity of rain, which aborted the coffee plants' flowers; hence, only 20% of the coffee plant could be harvested.

### 3.2. Farmers' Preferences on CSA Practices

Climate change affects coffee production and quality; however, temperature, humidity, and maximum precipitation changes will not be substantially impacted if effective cultivation techniques are implemented [47]. Therefore, several CSA practices have been developed to increase farmers' resilience and income. However, the application of CSA practices is very diverse because it is influenced by various factors such as the profile of farmers and the characteristics of the farming system.

Arabica farmers are an average of 48.4 years old, while Robusta farmers are an average of 40.8 years old. The average coffee farming experience in Arabica farmers group (8.80 years) is relatively lower than Robusta (14.09 years). Regarding the area and number of plants, the Arabica coffee area is relatively low, i.e., 0.83 ha with 1473 trees, while in the Robusta coffee area, it is 1.84 ha with 4279 trees (Table 2).

**Table 2.** Farming system characteristics of respondents.

| Aspects | Arabica Coffee | | | | | Robusta Coffee | | | | |
|---|---|---|---|---|---|---|---|---|---|---|
| | Min | Max | Mean | Std. Dev. | Description | Min | Max | Mean | Std. Dev. | Description |
| Land ownership (hectare) | 0.10 | 5.00 | 0.8360 | 0.77961 | | 0.50 | 32.00 | 1.8421 | 2.86198 | |
| Land status * | 1.00 | 4.00 | 1.4684 | 0.81391 | 1 = 72.2%; 2 = 10.1%; 3 = 16.5%; 4 = 1.3% | 1.00 | 5.00 | 1.1587 | 0.78397 | 1 = 96%; 5 = 4% |
| Number of trees (trees) | 50 | 12500 | 1472.72 | 1841.47 | | 150.00 | 12000.00 | 4278.97 | 2377.75 | |
| Mixed crops with annual crops ** | 0.00 | 1.00 | 0.8608 | 0.3484 | 0 = 13.9%; 1 = 86.1% | 0.00 | 1.00 | 0.1825 | 0.3878 | 0 = 81.7%; 1 = 18.3% |
| Mixed crops with perennial crops ** | 0.00 | 1.00 | 0.8734 | 0.3346 | 0 = 12.7%; 1 = 87.3% | 0.00 | 1.00 | 0.5000 | 0.5020 | 0 = 50%; 1 = 50% |
| Integrated coffee livestock ** | 0.00 | 1.00 | 0.8101 | 0.3947 | 0 = 19.0%; 1 = 81.0% | 0.00 | 1.00 | 0.2063 | 0.4063 | 0 = 79.4%; 1 = 20.6% |

Description: * 1 = owner; 2 = rent; 3 = partnership with private; 4 = partnership with local government; 5 = farm worker; ** 0 = not implemented yet; 1 = already implemented.

Based on land ownership status, 72.2% of Arabica farmers own land, 10.1% leased, 16.5% cooperate with the private sector, and 1.3% cooperate with the local government.

Most Robusta farmers manage their own land (96%). Based on farm diversification activities, most Arabica farmers integrate with Robusta farmers (and vice versa).

Implementing CSA practices includes integrated water, soil, and ecosystems management at a landscape scale [48]. Response mitigation includes emission reduction, sink enhancement, and fossil fuel offsetting, while adaptation includes technological development, adaptive farming practices, and financial management [49]. Financial management is related to income management, where incomes can be diversified by taking up non-farm income or implementing an integrated farming system. On the other side, various farmer practices represent local knowledge that can be used as a strategy to deal with extreme events and adapt to climate change [50].

In a more specific context, CSA practices in smallholder plantations consist of cultivation, soil management [51], water management, and farm diversification [52]. Climate-smart coffee practices are carried out, among others, by increasing coffee productivity through the intensification and application of coffee cultivation that is adaptive to climate change [53], as well as through the implementation of soil management, water management, and pest and disease management [54,55]. Coffee farming has seven functional groups: soil characteristics, water management, crop and genetic diversity, climate buffer and adjustment, crop nutrient management, structural elements and natural habitats, and system functioning [56]. Traditional agricultural techniques also play a role in climate change adaptation and mitigation because capacity is built based on agroecological characteristics [57].

Hence, in this study, the CSA practice that will be deepened is the practice applied by farmers, including cultivation, soil management, water management, and farm diversification. CSA practices include adopting climate-resilient varieties, changing planting dates and cropping patterns. Contour farming and balancing fertilisation are part of soil management, whereas water management includes *rorak* construction, reservoir development, and infiltration well construction. Diversification of the farming system entails integrating coffee with seasonal crops, perennial crops, and livestock. The description of CSA practices is shown in Table 3.

**Table 3.** CSA practices in smallholder coffee plantations.

| Attributes (CSA Aspect) | Levels (CSA Practices) | Description |
|---|---|---|
| Cultivation | Adopting of climate-resilient varieties | Use varieties or clones with improved morphology, yield, quality characteristics, resistance to pests and diseases, and environmental adaptability. |
| | Changing planting date | Adjustment of planting date due to availability of water or rainfall. |
| | Changing cropping patterns | Adjustment of cropping patterns from monoculture to polyculture. |
| Soil management | Contour farming | Practice of tillage, planting, and other farming operations performed on or near the contour of the field slope to promote positive row drainage and reduce ponding. |
| | Implementing balanced fertilisation | Balanced fertilization is the proper supply of all nutrients (macros and micros) throughout the growth of a crop. |

**Table 3.** *Cont.*

| Attributes (CSA Aspect) | Levels (CSA Practices) | Description |
|---|---|---|
| Water management | Making rorak | Rorak is a dead-end channel constructed next to the coffee plant to collect and absorb surface runoff water into the soil, slowing the runoff rate and placing organic fertilisers. |
| | Building reservoir | Water management with installation and setting up of irrigation techniques supported by technology in water-saving irrigation and distribution techniques. |
| | Making infiltration wells | Infiltration wells are rainwater conservation technologies widely applied to reduce surface runoff. |
| Farm diversification | Intercropping with seasonal crops | Coffee plants are cultivated simultaneously with seasonal crops in the same piece of land adhering to a specific row pattern |
| | Mixed-cropping with perennial crops | Coffee plants and other perennial crops are cultivated on the same piece of land simultaneously |
| | Implementing coffee livestock integration | Integrated coffee livestock systems are a sustainable intensification of agriculture that relies on synergistic relationships between plant and animal systems in a closed production cycle. |

There are four attributes and 11 levels of CSA. The attribute describes the CSA aspect of the coffee farming system, while the levels relate to the CSA practice. Through this design, a combination of attributes will be obtained. There are 9 (P1-9) stimuli generated from 4 attributes and 11 levels (Table 4).

**Table 4.** Stimuli in analysis.

| | Cultivation Aspect | Soil Management | Water Management | Diversification |
|---|---|---|---|---|
| P1 | Changing cropping patterns | Implementing balanced fertilisation | Building reservoir | Intercropping with seasonal crops |
| P2 | Changing cropping patterns | Contour farming | Making rorak | Mixed-cropping with perennial crops |
| P3 | Changing planting date | Contour farming | Building reservoir | Mixed-cropping with perennial crops |
| P4 | Changing planting date | Contour farming | Making infiltration wells | Intercropping with seasonal crops |
| P5 | Changing planting date | Implementing balanced fertilisation | Making rorak | Coffee livestock integration |
| P6 | Adopting climate-resilient varieties | Contour farming | Building reservoir | Coffee livestock integration |
| P7 | Adopting climate-resilient varieties | Contour farming | Making rorak | Intercropping with seasonal crops |
| P8 | Changing cropping patterns | Contour farming | Making infiltration wells | Coffee livestock integration |
| P9 | Adopting climate-resilient varieties | Implementing balanced fertilisation | Making infiltration wells | Mixed-cropping with perennial crops |

Based on the estimation results, the actual evaluation and assessment have a significant linear relationship for the Arabica and Robusta respondent groups. According to the statistical analysis, the Pearson's R and Kendall's tau correlation coefficients are fairly high,

at 0.996 and 0.986 for Arabica farmers and 0.992 and 0.986 for Robusta farmers, respectively, as well as statistically significant (Sig. < 0.05) (Table 5).

**Table 5.** Statistics analysis of farmers' preferences in CSA practices.

| Attributes | Arabica Farmers Importance Values | Robusta Farmers Importance Values | Levels | Arabica Farmers | | Robusta Farmers | |
|---|---|---|---|---|---|---|---|
| | | | | Utility Estimate | Std. Error | Utility Estimate | Std. Error |
| Cultivation technique | 26.315 | 22.736 | Adopting of climate-resilient varieties | 0.010 | 0.011 | 0.214 | 0.111 |
| | | | Changing planting date | 0.003 | 0.011 | −0.393 | 0.111 |
| | | | Changing cropping patterns | −0.013 | 0.011 | 0.179 | 0.111 |
| Soil management | 12.765 | 20.446 | Contour farming | −0.022 | 0.008 | 0.022 | 0.083 |
| | | | Implementing balanced fertilisation | 0.022 | 0.008 | −0.022 | 0.083 |
| Water management | 25.977 | 22.030 | Making rorak | 0.057 | 0.011 | −0.376 | 0.111 |
| | | | Building reservoir | −0.048 | 0.011 | 0.103 | 0.111 |
| | | | Making infiltration wells | −0.009 | 0.011 | 0.274 | 0.111 |
| Farm diversification | 34.943 * | 34.787 * | Intercropping with seasonal crops | 0.065 ** | 0.011 | −0.085 | 0.111 |
| | | | Mixed-cropping with perennial crops | 0.045 | 0.011 | 0.615 ** | 0.111 |
| | | | Implementing coffee livestock integration | −0.110 | 0.011 | −0.530 | 0.111 |
| | Pearson's R | | | 0.996 | 0.000 | 0.992 | 0.000 |
| | Kendall's tau | | | 0.986 | 0.000 | 0.986 | 0.000 |

* = attribute with the highest value; ** = level with the highest value.

The analysis revealed that farm diversification is the most important aspect in both regions, due to its potential benefits. Farm diversification is aligned with Integrated Farming Systems (IFS), which incorporate multiple farming systems within a specific geographic area and period. Arabica and Robusta farmers perceive diversification as a more desirable aspect of CSA than cultivation, soil management, and water management. Small-scale farmers will examine the length of the payback time when deciding on any CSA practice because agricultural production is characterised by high risks and low returns [58]. The decision to diversify agriculture activities is motivated by the potential economic benefits.

The IFS is divided into two components, i.e., the Mixed Crops System (MCS) and Integrated Crops Livestock System (ICLS). MCS is applied to seasonal and perennial crops, whereas ICLS is applied as coffee livestock integration. Arabica farmers prefer intercropping with annual crops, while Robusta farmers prefer intercropping with perennial crops.

Arabica farmers started as horticulturists, then they applied agroforestry in state forest areas through forest management programs with the community. Through community forest management programs, farmers in Bandung Regency practice coffee agroforestry in state forest lands. This collaboration is aligned with the local government's goal of revitalising the upstream area of the Citarum watershed. For Arabica farmers, land management is intended for conservation purposes, where seasonal crops are preferred in the transition period. Several crops were chosen, including large quantities of chilies and very small corn and tomatoes.

Farmers of Robusta cultivate coffee on their own initiative (96%). Land managed by farmers is individual land, where farmers are free to determine the development of coffee plantations. Robusta farmers prefer intercropping with perennial plants to increase their income. Some farmers in Robusta cultivate avocado, pepper, and dogfruit (*Pithecellobium lobatum Benth*).

The implementation of coffee livestock integration in Robusta farmers is relatively low, compared to Arabica farmers. Some Arabica farmers participate in the coffee livestock integration program launched by the local government. From an institutional perspective, both Arabica and Robusta farmers mostly are members of farmer groups and participate in extension activities, but only a few are members of cooperatives.

Cultivation techniques can be a preventive or mitigation step in dealing with climate variability or global warming in coffee plants [59]. In cultivation techniques, Arabica and Robusta smallholders tend to choose variety replacement. Varieties needed are resistant to drought and pests diseases, as well as support conservation agriculture. The new variety is expected to reduce the risk of product failure or quality degradation due to climate change [60].

Arabica farmers have distinct preferences, in terms of soil management, water management, and farm diversification, compared to Robusta farmers. From a soil management perspective, Arabica farmers frequently choose to contour farming, whereas Robusta farmers choose balanced fertilisation. Arabica farmers prefer infiltration wells from a water management standpoint, whereas Robusta farmers rorak making. Arabica farmers choose seasonal crop integration, whereas Robusta farmers prefer perennial crops.

The development of coffee plants for Arabica farmers is motivated by plant transfers to forest land for conservation purposes. Most farmers were originally horticultural farmers, where farming was carried out on the sloping ground. Farmers have been accustomed to planting land along contour lines to reduce erosion and runoff. Therefore, in land management, farmers tend to choose contour management.

Bandung Regency has hydrological potential in abundant water resources for both underground and surface water. Infiltration wells function as a place to collect rainwater and soak it into the ground. Farmers tend to choose infiltration wells to maintain surface flow and prevent flooding, while maintaining and increasing groundwater levels.

Robusta farmers are interested in applying balanced fertilization given the declining quality of the land. The framework for integrated soil fertility management has the potential to be critical in achieving sustainable intensification in CSA [61]. Farmers aspire to establish sustainable coffee crops by using balanced fertilising.

### 3.3. Barriers to Implementing CSA Practices

Adaptation to climate change reduces the negative effects of climate change or exploits the positive effects by making appropriate adjustments and changes [62]. Strategies include implementing technology or behavior change at the individual and community levels. However, farmers' preference for the CSA practises only shows the tendency of farmers to choose but does not mean that they will immediately proceed to the implementation decision stage. According to research conducted on New Zealand farmers, there is a disconnect between intentions and actual behavior regarding adopting climate change strategies [63].

The phenomenon of a gap between intention and actual behavior to adopt CSA practices is related to barriers faced. Smallholders face encouraging and inhibiting factors in CSA implementation. Barriers can exist on both the demand (user) and supply (technology provider) sides [23]. Multiple adoptions of innovations are explained mainly by access to critical resources (credit, income, and information), education level, and owned land size of the farmer [31]. The household size, monthly income, access to credit, and farmers' perception of climate change were all linked to the adoption of CSA practices [23]. Furthermore, access to and utilization of climatic information is required [48].

In-depth field observation revealed that coffee farmers faced two barriers: one linked to agricultural inputs and the other to the implementation process. The input barriers that coffee smallholders encounter when implementing CSA methods include a lack of experienced labor, capital, superior varieties, climate-smart cultivation techniques, weather information, and agricultural equipment and machinery. In addition, the process barriers faced by coffee smallholders are related to skills in dealing with climate disturbances, climate knowledge acquisition and use, access to agricultural extension, access to technical guidance, involvement in farmer groups and cooperatives, and the ability to obtain sustainable coffee production certification.

Arabica and Robusta farmers' perceptions regarding input barriers to CSA implementation in terms of skilled labor, capital, weather knowledge, and agricultural equipment and machinery significantly vary (Table 6).

**Table 6.** Input barriers to implementing CSA practices.

| | Skilled Workers | Capital | Superior Varieties | Climate-Smart Cultivation Technique | Climate Information | Agriculture Equipment and Machinery |
|---|---|---|---|---|---|---|
| Mann–Whitney U | 3361.000 | 4105.000 | 4709.000 | 4469.000 | 4248.000 | 3873.500 |
| Wilcoxon W | 6521.000 | 7265.000 | 12710.000 | 7629.000 | 7408.000 | 7033.500 |
| Z | −4.078 | −2.424 | −0.745 | −1.383 | −2.101 | −2.984 |
| Asymp. sig. (2-tailed) | 0.000 * | 0.015 * | 0.456 | 0.167 | 0.036 * | 0.003 * |

* significant at 5%.

Skilled workers are not a concern for Arabica farmers, with 39.24% strongly agreeing and 22.78% agreeing with the opinion (Figure 3). The location of Bandung Regency on the island of Java, is close to the center of province and national government, so there are many accessible workers with the necessary skills. Different conditions exist among Robusta farmers that live in rural places, where as much as 46.03% of respondents assessed that labor was a significant barrier to CSA implementation.

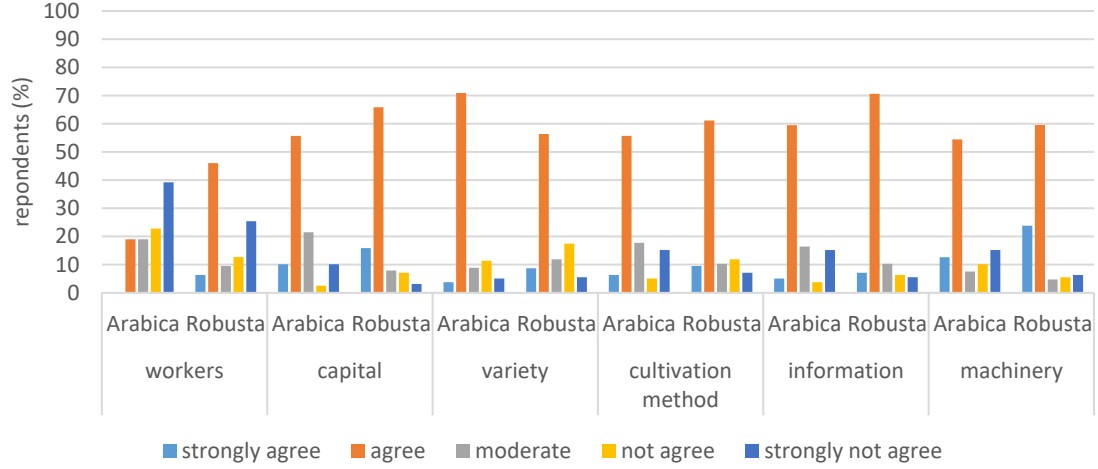

**Figure 3.** Distribution farmers' assessment of agriculture input barriers.

Robusta farmers stated that they agreed that elements such as capital (65.87%), climate data (70.63%), and agricultural equipment and machinery (59.52%) were barriers. Meanwhile, the percentage of Arabica farmers identifying these as barriers is lower. This condition is related to program support from the local government and technological support from research institutions for Arabica farmers. The local government has established a coffee development program for watershed restoration. Research institutes are assisting in developing an agricultural technology park in the Bandung Regency.

Arabica and Robusta farmers have different perceptions of the process barriers to CSA implementation, i.e., barriers related to developing skill development, knowledge transfer, getting access to agricultural extension services, getting technical assistance, being a part of farmer organisations, and obtaining certification for sustainable coffee production (Table 7).

**Table 7.** Process barriers to implementing CSA practices.

| | Skill Development | Knowledge Transfer | Agriculture Extension | Technical Assistance | Farmers Group Contribution | Partnership Involvement | Coffee Certification |
|---|---|---|---|---|---|---|---|
| Mann–Whitney U | 3826.000 | 3863.000 | 4069.500 | 3926.000 | 2880.000 | 4857.000 | 3870.000 |
| Wilcoxon W | 6986.000 | 7023.000 | 7229.500 | 7086.000 | 6040.000 | 8017.000 | 7030.000 |
| Z | −3.301 | −3.180 | −2.479 | −2.935 | −5.408 | −0.353 | −2.980 |
| Asymp. sig. (2-tailed) | 0.001 * | 0.001 * | 0.013 * | 0.003 * | 0.000 * | 0.724 | 0.003 * |

* significant at 5%.

Some Robusta farmers consider that a lack of competence, information, and assistance are significant barriers to CSA adoption, with a higher proportion of respondents strongly agreeing or agreeing than Arabica farmers (Figure 4). Although Arabica farmers have a lower average experience in coffee farming (8.80 years) than Robusta farmers (14.09 years), Arabica farmers receive some training on coffee cultivation as a CSA practice. This condition contrasts with Robusta farmers, who establish coffee plants on their initiative and capital.

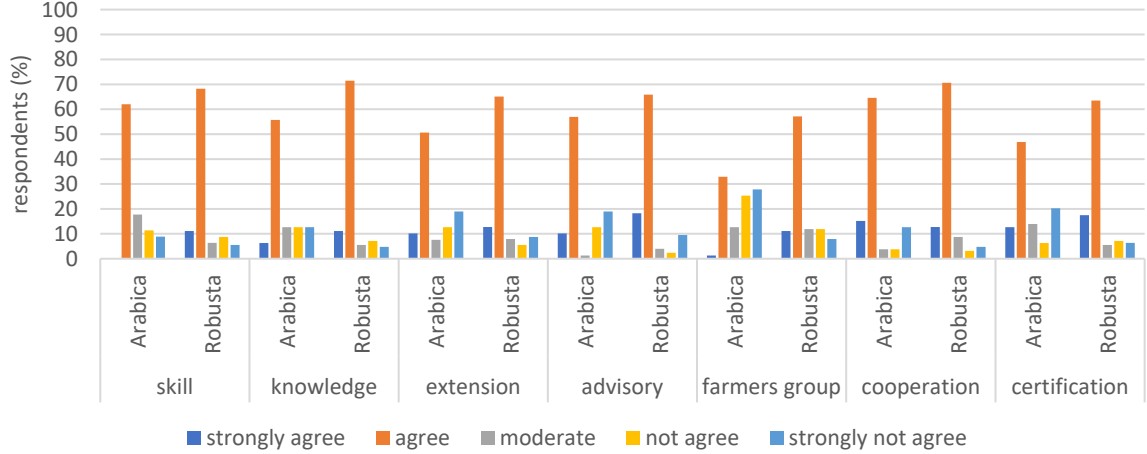

**Figure 4.** Distribution farmers' assessment of process barriers.

Cooperatives and farmer organisations can play a critical role in linking producers to markets [64]. Most Robusta farmers are not yet cooperative members and have not been active in the sustainable coffee certification. Coffee agribusiness is underdeveloped due to limited upstream and downstream integration, cooperation, financing, and marketing.

Both extension and credit support are somewhat constrained institutional aspects. Extension activities are not specific to coffee but focus on food crops. Agricultural financing schemes offered by formal financial institutions are extremely restricted and do not address coffee commodities specifically. All Arabica farmers are members of farmer groups, and the majority (68.35%) have participated in extension activities. Farmers use credit facilities at a proportion of 20.25% and cooperatives at a percentage of 15.19%. The same condition existed for Robusta farmers, where all farmers affiliate with farmer groups and the majority (53.97%) have participated in extension activities.

*3.4. Accelerating CSA implementation by Developing Agricultural Innovation Support System Model*

CSA can be described as an agricultural activity where the production process is adapted to climate change in parallel with being more ecological and maximised in food security [65]. Nonetheless, the impact of CSA is still far from being expected and, therefore, much debated because CSA adoption is possible if specific barriers can be overcome. Based on prior research into the challenges of implementing CSA practices, it is clear that the barriers are not simply technical issues at the individual farmer level but also non-technical issues at the ecosystem level. An integrative systems approach is required to understand how the various components of climate and agriculture connect, as well as how to balance these competing interests [66]. The collaboration of farmers, researchers, and extension workers is required to accelerate CSA adoption [67].

CSA practices refers to various activities, including behaviors, technology, climate information services, insurance, institutions, policies, and finance [68]. Technology influences farmers' preferences, readiness to adopt, and implementation cost [69]. Even among Arabica farmers who have benefited from multiple facilities and programs, some farmers confront similar constraints. The gender of the household head, size of the labor force, frequency of extension contacts, access to credit, access to weather forecasts, off-farm income, distance to input and output markets, number of traders, and asset ownership all have a significant effect on adoption intensity [70]. At the same time, a shortage of resources results in insufficient technical assistance to smallholder farmers, and a lack of collaboration limits program integration, different sources of innovation and knowledge, social learning, and sustainable CSA practices [71].

The proposed model, "Agricultural Innovation Support System", is an innovation-based support system that proportionally integrates input aspects in the form of "facility support" and process aspects in the form of "service support". The "Agricultural Innovation Support System" is built on "Agricultural Innovation Support Facilities", which provide technology and agricultural inputs, and "Agriculture Innovation Support Services", which provide services to develop a smallholder-based CSA ecosystem (Figure 5).

The development of an innovation-based agricultural system has created the Agricultural Knowledge System (AKS) model, which later transformed into the Agricultural Knowledge and Information System (AKIS) and Agricultural Innovation System (AIS). AKIS is a collection of agricultural actors engaged in the generation, transformation, transmission, storage, retrieval, integration, diffusion, and use of knowledge and information. In its application, the AKIS model is influenced by national institutions, laws, and cultures [72]. The Agricultural Innovation System (AIS) is a network of organisations, enterprises, and individuals focused on bringing new products, processes, and forms of organisation into economic use, together with the institutions and policies [73]. Specifically, the AIS model has been developed in various countries in the agricultural sector and certain commodities [74,75]. The implementation is affected by structural dimensions, namely the actors, interactions, and technologies, and varies by the strength and degree of integration [76].

Support facilities include innovative technology, smart farming, climate information systems, and financial resources. Innovative technology and smart farming utilise advances in information and communication technology to improve farm productivity, increase quality and yield, and reduce environmental footprint [77]. Access to agricultural credit and information will contribute to higher CSA adoption rates [78]. The significant variability of climate parameters experienced by farmers promotes the high demand for CIS [79]. Additionally, the availability of weather information has been implicated as a determinant of CSA practices [80].

Support services include capacity building, farmer organisations improvement, and network development support. The function of extension services in capacity building is more widely supported by farmers who are more likely to believe in climate change [81]. Membership of an agricultural association or group and the perception of the impact of climate change were statistically significant and positively correlated with the level of

CSA adoption [82]. Network development will promote cross-sector collaboration, policy coordination, and the participation of a diverse set of stakeholders as critical factors in the efficacy of CSA adoption [66].

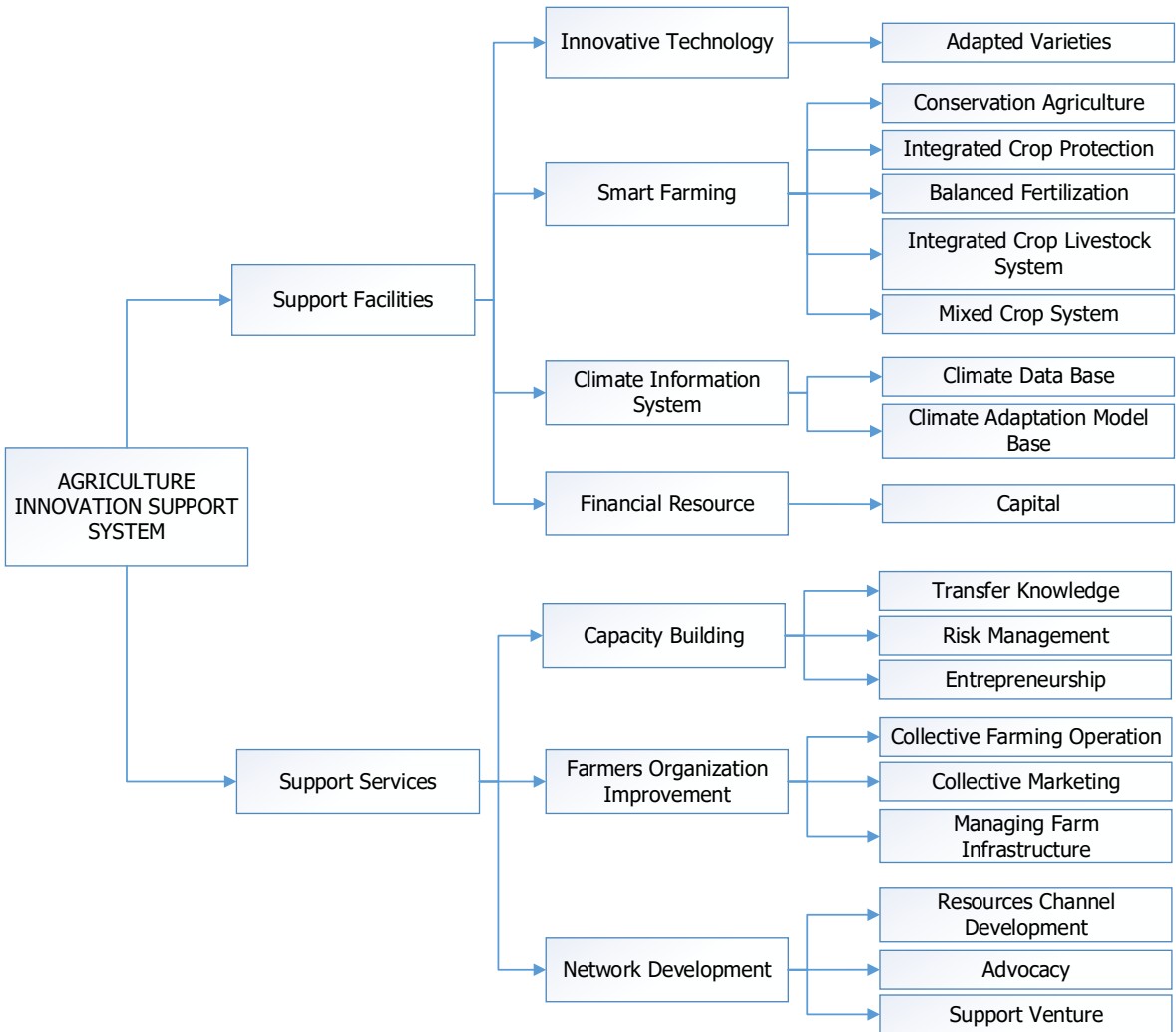

**Figure 5.** Proposed model of agricultural innovation support system.

The agricultural innovation support system model is designed to benefit smallholders by bringing together all segments of stakeholders in a framework including public sectors, private sector, universities, extension institutions, research institutions, and coffee associations. This model provides an overview of the elements needed to accelerate the adoption of CSA practices. However, the model usage should be complete with the information regarding the actors who must initiate or execute.

The agricultural innovation support system model can be operationalised by undertaking additional analysis, notably identifying priorities to develop policy recommendations. Pairwise comparisons were used to define priorities. According to an analysis of Arabica farmer groups, the suggested intervention aims to provide support services (0.800) by focusing on network development (0.432) (Figure 6). This recommendation is based on some existing programs and the level of technology adoption among Arabica farmers.

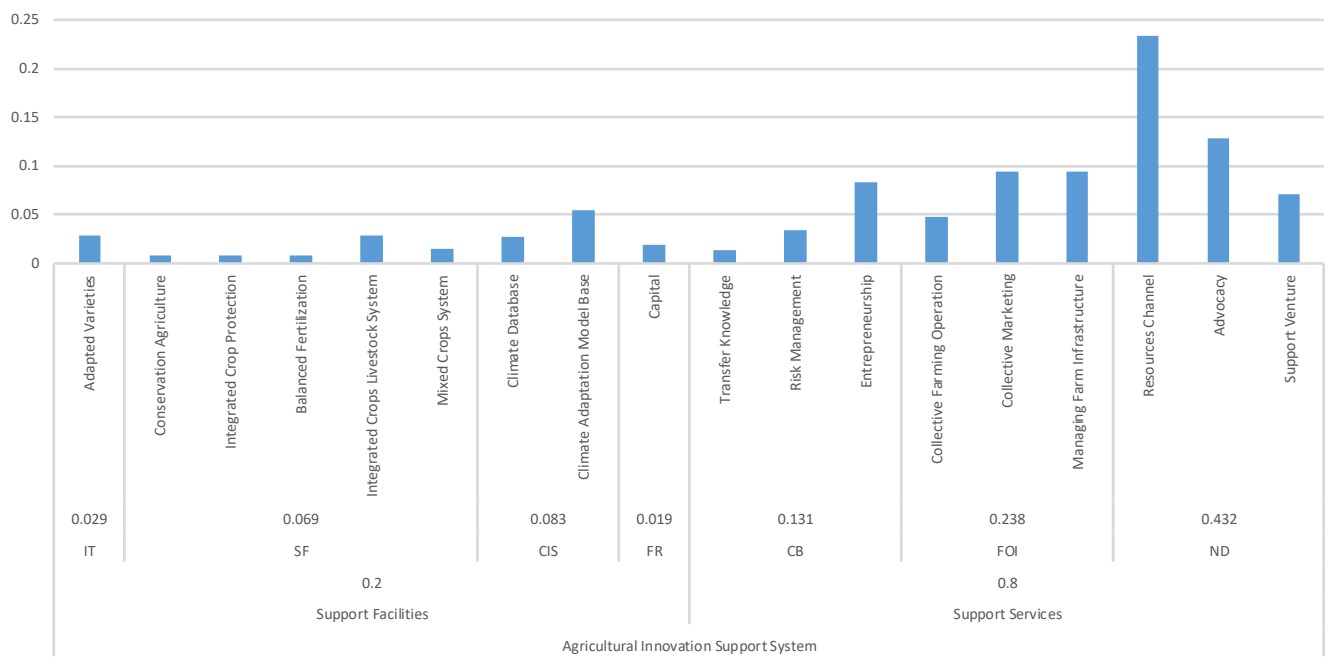

Description: IT = innovative technologies, SF = smart farming, CIS = climate information system, FR = financial resources, CB = capacity building; FOI = farmers organization improvement, ND = network development.

**Figure 6.** Priority number of agricultural innovation support system for Arabica farmers.

Based on the condition of the Robusta farmer group, intervention is directed at providing support facilities (0.750) and focusing on climate information system (0.311) (Figure 7). At the same time, the implementation of other smart farming practices (0.261) is also encouraged for increased crop production. The development of a climate information system is expected to promote the availability of climate data, enabling farmers to be more aware of climate change. The climate information system not only builds climate data in a platform but also completes it with climate information dissemination services to users. A climate information system can reduce climate vulnerability by managing the data, as well as enhancing information access, knowledge exchanges, and networks.

This study provides guidance for research institutions to accelerate technology adoption at the farmer level. For policymakers, a comprehensive picture will formulate an appropriate policy and operational program. Collective action and cross-sector coordination, which have been barriers [83], are projected to be overcome through the model's implementation. Farmers will benefit from innovation in various agricultural facilities, while also receiving guidance and assistance in adopting the technology to create resilient farmers and build sustainable coffee production.

It is envisaged that smallholders will adopt CSA practices if the technology is available and readily accessible to farmers by providing various support services. At the micro-level, CSA provides a guideline and some methods for restructuring and adapting farms to the sometimes extreme climatic conditions. Adoption of CSA practices positively and significantly improves farm net returns and reduces farmers' exposure to downside risks and crop failure [84]. At the macro-level, the massive implementation of CSA by smallholders will encourage economic development, poverty reduction, and food security by increasing agricultural productivity and income in a sustainable manner, as well as adapting and building resilience to climate change. A sustainable smallholder plantation-based coffee production system can be achieved by implementing a technology-based CSA and building an agricultural innovation support system that provides support facilities and support services.

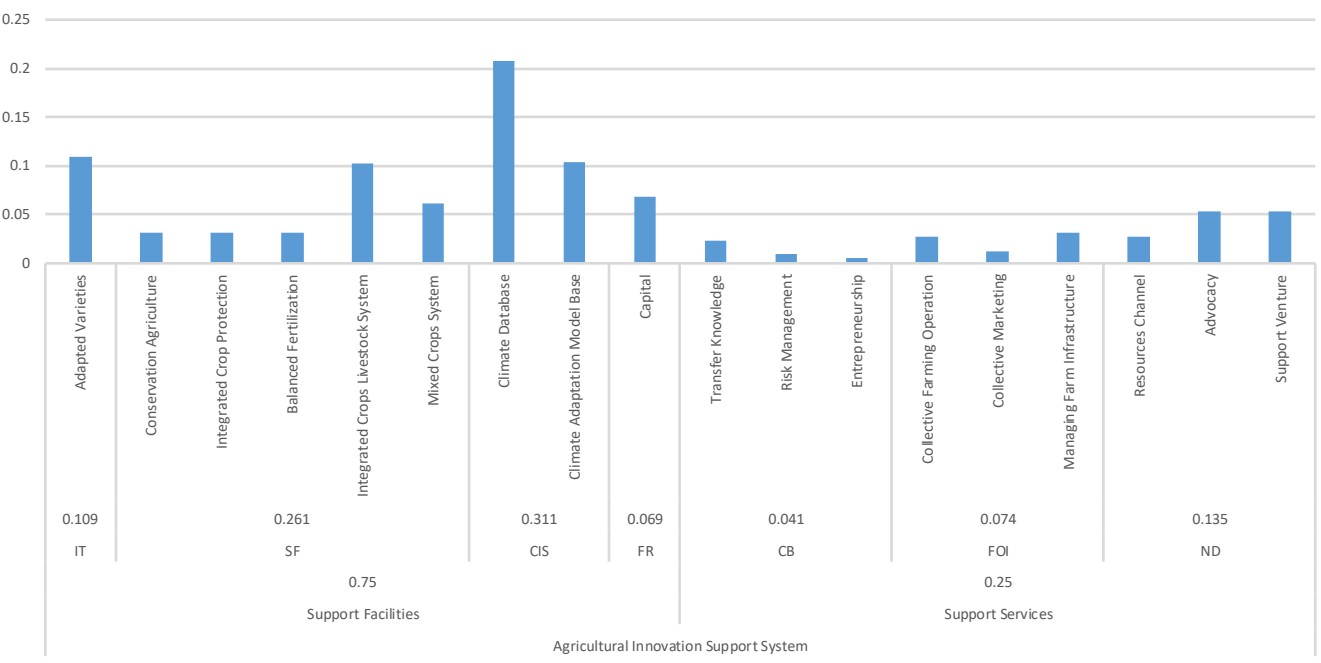

Description: IT = innovative technologies, SF = smart farming, CIS = climate information system, FR = financial resources, CB = capacity building; FOI = farmers organization improvement, ND = network development.

**Figure 7.** Priority number of agricultural innovation support system for Robusta farmers.

### 4. Conclusions

Climate change, which significantly impacts the production and quality of Arabica and Robusta coffee, imposes an additional burden on smallholders, the primary actors. Several CSA practices have been identified, which include cultivation, soil management, water management, and farm diversification. Farm diversification is a more desirable aspect of CSA to Arabica and Robusta farmers. In more detail, Arabica farmers prefer intercropping with annual crops, whereas Robusta farmers prefer perennial crops. Farmers' preferences for CSA practices are influenced by factors such as farmer characteristics, agricultural system profiles, agroecological conditions, reasons for planting, and local government programs and policies.

The farmers' preferences and availability of technology do not automatically lead to CSA practices adoption at the farmer level. Arabica and Robusta farmers are challenged with driving and inhibiting factors in implementing CSA. Coffee farmers face two types of barriers: those connected to agriculture inputs and those linked to the implementation process. The agriculture input barriers include a lack of experienced labor, capital, superior varieties, climate-smart cultivation techniques, climate information, and agricultural equipment and machinery. In addition, the implementation process barriers are related to skills in dealing with climate change, knowledge acquisition and use, agricultural extension access, technical guidance, involvement in farmer organisations and cooperative, and the ability to get sustainable coffee production certification.

This paper proposes a model of an agricultural innovation support system as an innovation-based support system. The model provides guidance for designing strategies to overcome numerous technological and institutional barriers that individual farmers cannot manage. This strategy comprises two components: (1) agricultural innovation support facility, which offers agriculture technology and inputs, and (2) agricultural innovation support services, which provide services to facilitate the development of a farmer-based CSA ecosystem.

Strategy can be constructed by considering the components of the agricultural innovation support system model. The priority intervention for Arabica farmers is to provide

support services with an emphasis on network development, taking into account the production system that has been established and the climate awareness that has been built. Providing support facilities focusing on developing a climate information system to increase knowledge of climate change and CSA practices is the priority intervention for Robusta farmers. It is envisaged that the establishment of a systems-based strategy can accelerate the implementation of the CSA.

**Author Contributions:** Conceptualisation, F.D. and S.W.; methodology, S.W. and R.V.; formal analysis, F.D., S.W. and R.V.; resources, F.D.; writing—original draft preparation, F.D., S.W. and R.V.; writing—review and editing, S.W. and R.V.; supervision, F.D.; funding acquisition, F.D. All authors have read and agreed to the published version of the manuscript.

**Funding:** This research received no external funding.

**Conflicts of Interest:** The authors declare no conflict of interest.

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
