# Peer review of "Climate Smart Agriculture Implementation on Coffee Smallholders in Indonesia and Strategy to Accelerate"

_land, doi:10.3390/land11071112_

Round 1

Reviewer 1 Report

The study "Developing Agricultural Innovation Support System to Accelerate Climate Smart Agriculture Implementation for Coffee Smallholders in Indonesia. " aims to (1) analyze farmers' preferences towards CSA as strategies for climate change adaptation at Arabica (Coffea arabica) and Robusta coffee (Coffea canephora) production centers in Indonesia, (2) analyze barriers to implementing CSA at the coffee farmer's level, and (3) provide a model agricultural innovation support system to accelerate the implementation of CSA for coffee smallholders.

 Comments and suggestions:

Overall the work quality is good; the research objectives are archives. The study is in the Land journal's scope and can be considered for publication after some modifications.

·       Line 31-34. For the information presented there, it's better to follow the latest IPCC report AR6 and update it accordingly.

·       Line 36-37. "with global warming of 1.5°C and increase further with 2°C." Recheck with AR5 or AR6.

·       Line 64. "External factors," e.g. ???

·     Line 99-107. Information presented there more fits with the conclusion instead of here.

·       Section 2. Study area details should be elaborate clearly. It's better to provide a detailed study area map to present the location mentioned in table 1.

·       Section 2.2. I have a bit of doubt about the selection of the respondents. Also, it's better to report the questionnaire. Furthermore, add the reference for the secondary data.

·       Line 124-127. Consider citing properly to validate the utilized methods.

·       Line 164. Describe Kendall's Tau correlation and Mann-Whitney rank test

·       Line 174. Describe Hypothesis H0 and H1 clearly, then describe "If the significance value> 0.05, then H0 is accepted and H1 is rejected".

·       It's better to add to a detailed methodology flowchart.

  ·       Line 495. It's better to provide the rules and limitations for the proposed model, "Agricultural Innovation Support System,"

·       Also, discuss research results clearly instead of mixing them with the results section, and provide further details about the limitations of this work and future implications and recommendations. 

·       The conclusion of this work is so extended. Make it more comprehensive.

·       In the reference section, the DOI of all citations is missing. 

Author Response

We would like to thank you for giving us the opportunity to submit a revised draft of the manuscript for publication in the “Land”. We appreciate the time and effort that the reviewers dedicated to providing feedback on our manuscript and are grateful for the insightful comments on and valuable improvements to our paper. We have incorporated most of the suggestions made by the reviewers. Those changes are highlighted within the manuscript. Please see below, in blue, for a point-by-point response to the reviewers’ comments and concerns. All page numbers refer to the revised manuscript file with tracked changes.

Reviewer 2 Report

This is an interesting and well-performed study on the specifics of small-size coffee farming in Indonesia and the attitudes of local farmers to innovations and support services. 

In general, I think the manuscript is acceptable. However, I would suggest the author consider the following issues:

How were the respondents selected? Please describe the approach to selecting the respondents and building the array in more detail. What about the representation of the array?

More land-related characteristics of the respondents are required aside the specialization in a type of coffee and the location - for instance, size and differences in sizes. In the title, the author refers to smallholders. How small are they? Do their sizes differ? What are the average size and the extremes? Does the array adequately reflect the differences in sizes and represent all types of land holders? Table 2 breaks the holders into categories, but what about the initial array presented in Table 1?

The number of respondents - 202 (line 114) or 205 (table 1)?

A map of the territory would be helpful. 

The data analysis methods should be discussed in a critical manner. The author should convince a reader that the methods used in the paper are relevant and best serve the aim and specifics of the study. What are the advantages and disadvantages of these particular methods in relation to the aim of the current study? Have they been used in other studies? If yes, what were the results? Are there alternative approaches? What are their characteristics in relation to the selected methods? 

Make sure the text is proofread by a native speaker of English to improve the language and style. 

Author Response

(The authors gave the same response as above.)

Reviewer 3 Report

Title – it does not properly describe the content of the paper;

Abstract – information about the methodology used for data collection and the main policy implications are missing.

Introduction:

-          Line 55 to 56 – what means small coffee holders’ plantation in Indonesia?  A comparison with the other important coffee producers will help the reader to better understand the issue of CSA in the context of Indonesia;

-          In the introduction part the authors fail to present the existing public or private measures to support CSA in Indonesia and in the world;

Material and methods

-          Not clear how the number of the respondents from table 1 was established. Are there any statistical methods used to calculate that numbers?

-          The questionnaire content is not properly described; The types of statistical secondary data are not presented;

Results

-          Section 3.1 is more appropriate from the structural point of view for introduction or material sections;

-          Table 2 is a descriptive statistic of the questionnaire results; 0 versus 1 from the description section is not clear how to be understand;

-          Table 3 present a list of attributes for the CSA. It is not clear how this list was created because the introduction and the method sections fails to present in detail other studies that worked with CSA attributes;

-          Table 4 deals with methodological issues;

Discussion

-          Fails to discuss the main findings in the context of other previous studies;

-          There is no link between the proposed model of Agricultural Innovation and Support System (figure 4) and other previous studies. To understand that model it necessary to described in detail the current Agricultural Knowledge and Innovation System (AKIS) from the Indonesian coffee sector and to linked it with the current situation from other important coffee producers in the world.

Author Response

(The authors gave the same response as above.)

Round 2

Reviewer 1 Report

The authors have addressed all my suggestions, and the manuscript is now significantly improved and acceptable for publication.

Reviewer 3 Report

The authors properly responded to the main critics pointed out in the first review process.